# Comparison between Conventional Smear and Liquid-Based Preparation in Endoscopic Ultrasonography-Fine Needle Aspiration Cytology of Pancreatic Lesions

**DOI:** 10.3390/diagnostics10050293

**Published:** 2020-05-09

**Authors:** Soo Hee Ko, Jung-Soo Pyo, Byoung Kwan Son, Hyo Young Lee, Il Whan Oh, Kwang Hyun Chung

**Affiliations:** 1Eulji University School of Medicine, Daejeon 34824, Korea; soohee8803@naver.com; 2Department of Pathology, Daejeon Eulji University Hospital, Eulji University School of Medicine, Daejeon 35233, Korea; jspyo@eulji.ac.kr; 3Department of Internal Medicine, Nowon Eulji Hospital, Eulji University School of Medicine, Seoul 01380, Korea; 2hyo0@eulji.ac.kr (H.Y.L.); 20180121@eulji.ac.kr (I.W.O.); kh.chung@eulji.ac.kr (K.H.C.)

**Keywords:** pancreas, endoscopic ultrasonography-fine needle aspiration cytology, conventional smear, liquid-based preparation, meta-analysis, diagnostic test accuracy review

## Abstract

The present study aimed to compare the diagnostic accuracy between conventional smear (CS) and liquid-based preparation (LBP) in endoscopic ultrasonography-fine needle aspiration cytology (EUS-FNAC) of pancreatic lesions. Using 31 eligible studies, the diagnostic accuracy of cytologic examination in CS and LBP was evaluated through a conventional meta-analysis and diagnostic test accuracy review. Overall concordance rates were 82.8% (95% confidence interval [CI], 79.8–85.5%) and 94.0% (95% CI, 84.4–97.8%) in CS and LBP, respectively. CS with rapid on-site evaluation (ROSE) showed a higher concordance rate than CS without ROSE. In CS, the pooled sensitivity and specificity were 89.8% (95% CI, 85.2–93.1%) and 95.0% (95% CI, 90.0–97.6%), respectively. The diagnostic odds ratio (OR) and area under curve (AUC) of the summary receiver operating characteristic (SROC) curve were 90.32 (95% CI, 43.85–147.11) and 0.945, respectively. In LBP, the pooled sensitivity and specificity were 80.9% (95% CI, 69.7–88.7%) and 99.9% (95% CI, 1.5–100.0%), respectively. The diagnostic OR and AUC of the SROC curve were 57.21 (95% CI, 23.61–138.64) and 0.939, respectively. Higher concordance rates were found in CS with ROSE and LBP in EUS-FNAC of pancreatic lesions. Regardless of the cytologic preparation method, EUS-FNAC is a useful and accurate diagnostic tool for pancreatic lesions.

## 1. Introduction

Fine needle aspiration (FNA) using endoscopic ultrasonography (EUS) was introduced in the 1990s and has become a mainstay in recent years [1,2]. Pancreatic lesions include not only solid masses but also cystic lesions. Sampling from cystic lesions may not be as effective as sampling from solid lesions. In addition, sampling from pancreatic lesions has some limitations due to their anatomical location. To improve the diagnostic yield of pancreatic lesions, various protocol variations on EUS-FNA equipment and techniques have been studied and applied [1,2]. Due to advances in EUS techniques and the development of various tissue acquisition instruments, the diagnostic accuracy using EUS-FNA has been substantial. In addition, various cytologic preparation methods have been developed and applied in the diagnostic field. For cytologic diagnosis, cytologic samples for EUS-FNA are prepared using a conventional smear (CS) or liquid-based preparation (LBP). LBP methods were developed in the 1990s and have been widely utilized in the uterine cervix [3,4]. The quality of samples and effectiveness of LBP have been improved in non-gynecologic organs, including pancreatic lesions [5,6,7,8]. However, CS is still used in many pathologic laboratories. Diagnostic yields of EUS-fine needle aspiration cytology (FNAC) have been reported [1,2,9,10,11,12,13,14,15,16,17,18,19,20,21,22,23,24,25,26,27,28,29,30,31,32,33,34,35,36,37], but detailed information regarding the cytologic preparation method is not available. EUS-FNA was associated with higher diagnostic sensitivity and specificity for pancreatic cancer [38,39]. However, various factors, including the cytologic preparation method, can affect the diagnostic accuracy of EUS-FNA [40,41,42]. Since each study obtained data from different EUS-FNAC methods, a meta-analysis is suitable for detailed comparisons.

To elucidate the diagnostic accuracy of CS and LBP in EUS-FNAC of pancreatic lesions, a conventional meta-analysis and diagnostic test accuracy (DTA) review were performed. In addition, sample adequacy from CS and LBP in EUS-FNAC of pancreatic lesions was investigated.

## 2. Materials and Methods

### 2.1. Published Study Search and Selection Criteria

Relevant articles were obtained by searching the PubMed databases through 15 November 2018. This database was searched using the following keywords: “(pancreas or pancreatic) and (endoscopic ultrasound or endosonography or EUS) and (fine-needle or fine-needle aspiration or fine-needle biopsy).” The titles and the abstracts of all searched articles were screened for exclusion. Review articles and previous meta-analysis were also screened to obtain additional eligible studies. Searched results were then reviewed and articles were included if the study investigated the pancreatic lesions and there was information for the CS and LBP. The articles that were case reports, non-original articles, or non-English language publications were excluded.

### 2.2. Data Extraction

Data from all eligible studies were extracted by two individual authors. Extracted data from each of the eligible studies included the following [1,2,9,10,11,12,13,14,15,16,17,18,19,20,21,22,23,24,25,26,27,28,29,30,31,32,33,34,35,36,37]: first author’s name, year of publication, study location, number of patients analyzed, type of pancreatic lesions, the methodology of cytologic preparation, the presence of rapid-on site cytologic examination (ROSE), and needle size. For the meta-analysis, we extracted all data associated with the diagnostic accuracy of CS and LBP in EUS-FNAC of pancreatic lesions. In addition, numbers of true positive, false positive, false negative, and true negative of each cytologic preparation were investigated to obtain the sensitivity, specificity, diagnostic odds ratio, and the summary receiver operating characteristic (SROC) curve.

### 2.3. Statistical Analysis

To obtain the diagnostic accuracy of CS and LBP, a meta-analysis was performed using the Comprehensive Meta-Analysis software package. (Biostat, Englewood, NJ, USA). Diagnostic accuracy was evaluated by concordance between EUS-FNAC and histologic diagnosis. As the eligible studies used various methods for pancreatic lesions and had different numbers of patients, a random-effects model was more appropriate than a fixed-effects model. Heterogeneity between the eligible studies was checked using *p* statistics (*p*-value). To evaluate publication bias, Begg’s funnel plot and Egger’s test were conducted. The results with *p* < 0.05 were considered statistically significant. If significant publication bias was found, the fail-safe N and trim-fill tests were additionally conducted to confirm the degree of publication bias. The results were considered statistically significant with *p* < 0.05.

The diagnostic test accuracy (DTA) review of CS and LBP in EUS-FNAC was performed using R software ver. 3.6.3. We calculated the pooled sensitivity and specificity, the diagnostic odds ratio (OR) according to individual data was collected from each eligible study in various categories of comparison. By plotting the ‘sensitivity’ and ‘1-specificity’ of each study, the SROC curve was constructed first and the curve fitting was performed through linear regression. As each dataset was heterogeneous, the accuracy data were pooled by fitting a SROC curve and measuring the value of the area under the curve (AUC). An AUC close to 1 means the test is strong and an AUC close to 0.5 means the test is considered poor. According to the cytologic preparation method, ROSE or not, needle size, and types of pancreatic lesions, subgroup analysis was conducted.

## 3. Results

### 3.1. Selection and Characteristics

A total of 2557 studies were identified through database searching. Due to insufficient information on concordance rates and diagnostic accuracy, 1265 studies were excluded. An additional 908 studies were excluded because they were not original studies, 335 were excluded as they studied other diseases, 13 were excluded as they were not in English, 4 were excluded because they were non-human studies, and 1 was excluded as it was duplicated research. Finally, 31 studies were included in the present meta-analysis (Figure 1 and Table 1), providing data from 5776 patients. Detailed information of eligible studies is shown in Table 1.

### 3.2. Comparison of Sample Adequacy between Conventional Smear and Liquid-Based Preparation

The sample adequacies of CS and LBP were 0.955 (95% confidence interval (CI), 0.923–0.974) and 0.938 (95% CI, 0.801–0.983), respectively. The sample adequacies of CS with and without ROSE were 0.953 (95% CI, 0.898–0.979) and 0.947 (95% CI, 0.484–0.997), respectively.

### 3.3. Comparison of Diagnostic Accuracy between Conventional Smear and Liquid-Based Preparation

The diagnostic accuracies of CS and LBP, investigated using the concordance rate between cytologic and histologic diagnoses, were 0.828 (95% CI, 0.798–0.855) and 0.940 (95% CI, 0.844–0.978), respectively (Table 2). In CS, the estimated values of diagnostic accuracy of pancreatic solid and cystic lesions were 0.824 (95% CI, 0.792–0.852) and 0.800 (95% CI, 0.572–0.923), respectively. CS with ROSE had a higher diagnostic accuracy than CS without ROSE (0.928, 95% CI, 0.879–0.959 vs. 0.809, 95% CI, 0.748–0.858). Needle size had no effect on the diagnostic accuracy of EUS-FNAC (0.808, 95% CI, 0.682–0.892 and 0.808, 95% CI, 0.720–0.873 in 22- and 25-gauge, respectively). However, the diagnostic accuracy of LBP with a 22-gauge was higher than that of LBP with 25-gauge (0.983, 95% CI, 0.935–0.996 vs. 0.902, 95% CI, 0.844–0.940). In CS, the diagnostic accuracies of puncture methods were 0.762 (95% CI, 0.577–0.882) and 0.588 (95% CI, 0.487–0.681) in slow-pull and fanning techniques, respectively. To assess publication bias, Begg’s funnel plot and Egger’s test were preferentially conducted. In CS without ROSE, significant publication bias was identified in the primary assessment. To define the degree of publication bias, the fail-safe N test and the trim and fill tests were conducted as the secondary assessment, and no significant publication bias was found. In the assessments of other subgroups, no significant publication biases emerged.

### 3.4. Diagnostic Test Accuracy Review of Cytology

A DTA review was conducted to elucidate the DTA of CS and LBP in EUS-FNAC of pancreatic lesions. The pooled sensitivities of CS and LBP were 0.90 (95% CI, 0.85–0.93) and 0.81 (95% CI, 0.70–0.89), respectively (Figure 2 and Figure 3). The pooled specificities were 0.95 (95% CI, 0.90–0.98) and 1.00 (95% CI, 0.02–1.00), respectively. The AUC of SROC was slightly higher in CS than LBP (0.945 vs. 0.939). The sensitivity was higher in CS with ROSE than in CS without ROSE (0.93, 95% CI, 0.88–0.96 vs. 0.84, 95% CI, 0.67–0.93). The diagnostic OR and AUC of SROC was higher in CS with ROSE than in CS without ROSE (diagnostic OR: 102.50, 95% CI, 39.24–267.72 vs. 24.42, 95% CI 8.11–73.47 and AUC of SROC: 0.952 vs. 0.884).

## 4. Discussion

EUS-FNAC has been introduced and has emerged as a safe and accurate technique in the diagnosis of pancreatic lesions [43]. Many studies on the diagnostic accuracy and effectiveness of EUS-FNAC have been reported [1,2,9,10,11,12,13,14,15,16,17,18,19,20,21,22,23,24,25,26,27,28,29,30,31,32,33,34,35,36,37]. Cytologic preparation is divided into CS and LBP in daily practice [3]. Although LBP has advantages in cytologic preparation and diagnosis, CS is widely used in daily practice. The diagnostic accuracy of CS and LBP is controversial [25,26,29,44]. To the best of our knowledge, the present study is the first meta-analysis and DTA review to compare EUS-FNAC of pancreatic lesions with CS and LBP.

LBP usage has gradually increased and replaced CS. LBP has been widely applied for screening tests of the uterine cervix and non-gynecologic examination in daily practice. Results comparing CS and LBP in EUS-FNAC were reported in a previous literature review [3]. However, conclusive information is not available owing to heterogeneous results of previous studies [25,26,29,44]. Specifically, the sensitivities of CS and LBP ranged from 61.7% to 91.0% and from 58.0% to 97.9%, respectively. The specificities of CS and LBP were both 100%. Therefore, a meta-analysis is needed in order to clarify the diagnostic accuracy of CS and LBP. In our results, the diagnostic accuracies of CS and LBP were 0.828 (95% CI, 0.798–0.855) and 0.940 (95% CI, 0.844–0.978), respectively. Based on diagnostic accuracy, LBP might be a superior method to CS in EUS-FNAC of the pancreatic lesion. In the DTA review, the sensitivity was higher in CS, but the specificity was higher in LBP. In addition, considering the diagnostic OR and AUC of SROC, CS may be a more accurate method than LBP. However, the diagnostic OR and AUC of SROC of CS without ROSE were lower than those of CS with ROSE or those of LBP. Thus, in EUS-FNAC with CS, the impact of ROSE on diagnostic accuracy should be considered. These results suggest that the application of ROSE is more important in CS of EUS-FNAC. The advantages of LBP are the usage of an automated method and reduction of the false-negative rate [3]. In addition, further evaluation including genetic tests and immunocytochemistry is possible, unlike with CS.

The diagnostic accuracy of EUS-FNAC for pancreatic lesions can be affected by several factors, including the characteristics of the pancreatic lesion itself, such as its characterization as the cystic or solid type. The cytologic diagnosis of pancreatic cystic lesions is difficult because various diseases may be associated with them, such as pseudocysts, mucinous cystadenoma with or without invasive carcinoma, intraductal papillary mucinous neoplasm, and serous adenocarcinoma. In the previous meta-analysis, the pooled sensitivities were 0.51 to 0.52 and the pooled specificities were 0.94 to 0.97 in pancreatic cystic lesions [45]. Thornton et al. reported that the sensitivity and specificity were 0.54 and 0.93, respectively [46]. However, the previous studies evaluated the diagnostic accuracy of EUS-FNAC without differentiation between CS and LBP in pancreatic cystic lesions. Thus, the comparison of DTA between CS and LBP in pancreatic cystic lesions is needed. However, the detailed comparison could be not performed owing to insufficient information about eligible studies. In the present study, the overall diagnostic accuracy of CS was 0.800 (95% CI, 0.572–0.923) in pancreatic cystic lesions. There was no significant difference in diagnostic accuracy between pancreatic solid and cystic lesions (0.824 vs. 0.800). Although the sensitivity of cytologic examination might be lower in cystic lesions, the diagnostic accuracy was not lower than that in solid lesions. As described above, because of the application of ancillary tests, LBP is more suitable in the assessment of pancreatic cystic lesions.

ROSE may be more useful in CS because the CS method is not reproducible. Previously, the role of ROSE was reported using a meta-analysis [39]. The sensitivity was higher in EUS-FNAC with ROSE than in EUS-FNAC without ROSE [39]. There was no difference in specificity [39]. However, the comparison between CS and LBP was not performed in the previous meta-analysis. In our results, CS with ROSE resulted in a higher diagnostic accuracy than CS without ROSE (0.928, 95% CI, 0.792–0.852 vs. 0.748, 95% CI, 0.748–0.858). In the DTA review, CS with ROSE had higher DTA than CS without ROSE. As a result, in EUS-FNAC using CS, the application of ROSE may be useful for increasing the diagnostic accuracy of EUS-FNA. However, in LBP, the application of ROSE has no impact on the diagnostic accuracy of EUS-FNAC (0.980, 95% CI, 0.871–0.997 vs. 0.983, 95% CI, 0.888–0.998). For EUS-FNAC with ROSE, the assessment of sample adequacy is done using the smear method with fast staining. In the meta-analysis, there was no significant difference in sample adequacy between CS with and without ROSE (0.953, 95% CI, 0.898–0.979 vs. 0.947, 95% CI, 0.484–0.997). After adequate sampling, the cytologic preparation is selected between CS or LBP. As ROSE is based on the smear method with fast staining, the impact of ROSE might be lower in LBP than in CS. Applying ROSE in LBP can be considered to evaluate both LBP and CS slides. Thus, because this cannot be a pure LBP, it is unable to evaluate an impact on diagnostic ability of ROSE in LBP. In addition, an experienced pathologist or cytotechnician should conduct ROSE. Elucidating the role of ROSE will require direct comparison between slides from ROSE and the actual process.

There are a number of limitations in the current study. First, the detailed comparison between pancreatic solid and cystic lesions could be not conducted in LBP samples, owing to insufficient information from eligible studies. Second, the investigation of diagnostic accuracy from various LBP methods could be not conducted, due to insufficient information from eligible studies. Third, since cytologic examinations can produce an additional cell block, the detailed comparison between LBP with cell block and CS will be required. Fourth, detailed evaluations based on different diseases of pancreatic cystic lesions could be not performed. Fifth, the pooled sensitivities of 22- and 25-gauges of CS were 0.89 (95% CI, 0.88–0.91) and 0.78 (95% CI, 0.71–0.84), respectively. The pooled specificities were 0.93 (95% CI, 0.89–0.95) and 1.00 (95% CI, 0.87–1.00), respectively. The diagnostic ORs were 91.24 (95% CI, 36.88–225.75) and 120.79 (95% CI, 15.23–957.83) in 22- and 25-gauges of CS, respectively (data not shown). However, a DTA review for comparison between 22- and 25-gauges in LBP could not be performed due to insufficient information. Sixth, the subgroup analysis based on puncture needle types could not be performed due to insufficient information.

## 5. Conclusions

In conclusion, our data show that EUS-FNAC is a useful diagnostic tool for pancreatic lesions, regardless of the cytologic preparation method. Among cytologic preparation methods, CS with ROSE and LBP have the highest diagnostic accuracy in EUS-FNAC of pancreatic lesions. In addition, because ROSE can be useful for increasing diagnostic accuracy, the recommendation of ROSE is necessary for EUS-FNAC using CS.

## Figures and Tables

**Figure 1 diagnostics-10-00293-f001:**
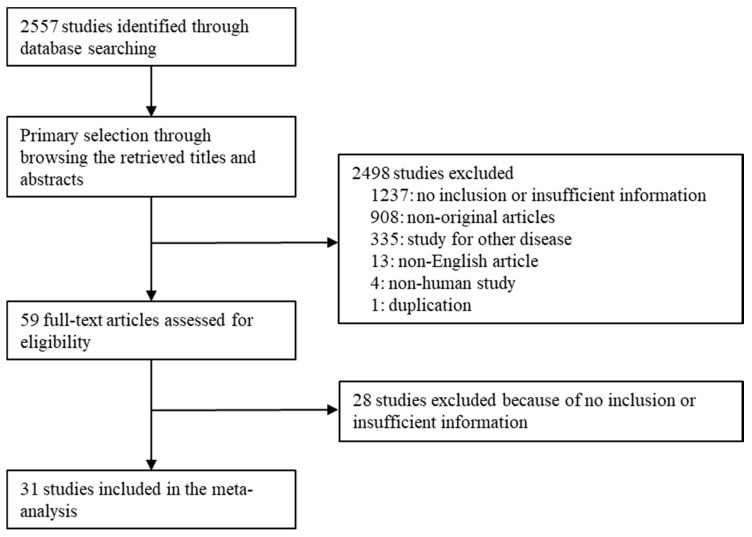
Flow chart for study search and selection methods.

**Figure 2 diagnostics-10-00293-f002:**
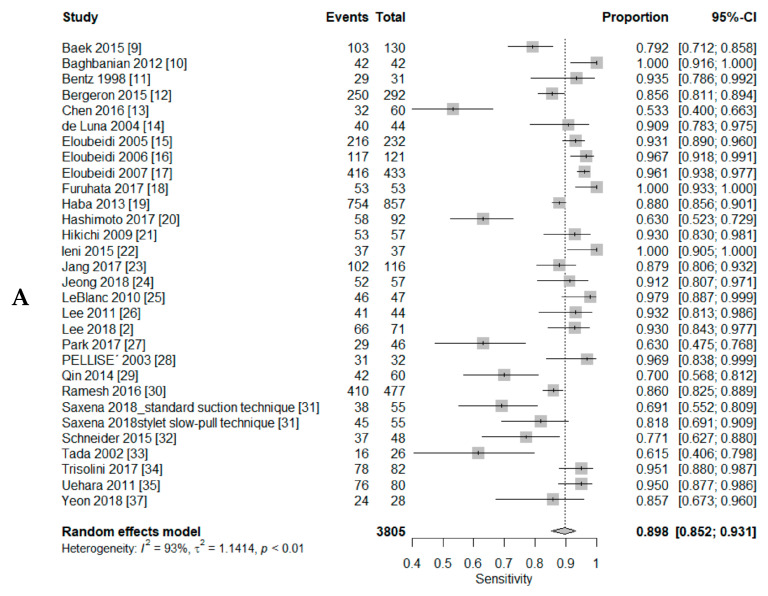
Forest plot diagram of the pooled sensitivit y (**A**), specificity (**B**), and summary receiver operating characteristic (SROC) curve (**C**) in the conventional smear. (Triangle, estimate of each study)

**Figure 3 diagnostics-10-00293-f003:**
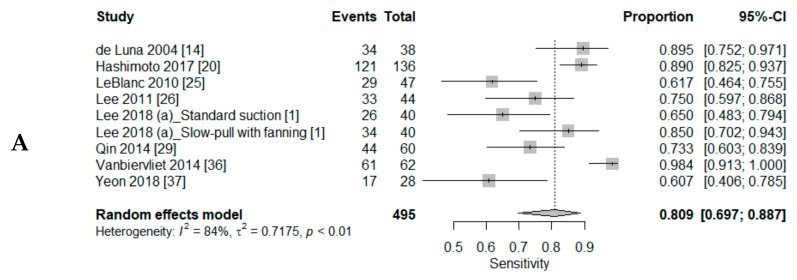
Forest plot diagram of the pooled sensitivity (**A**), specificity (**B**), and summary receiver operating characteristic (SROC) curve (**C**) in liquid-based preparation. (Triangle, estimate of each study)

**Table 1 diagnostics-10-00293-t001:** Main characteristics of the eligible studies.

Study	Cell Preparation		Type of Lesion	No of Patients or Cases	Needle Size	ROSE
Baek, 2015 [9]	CS		PSM	191	ND	ND
Baghbanian, 2012 [10]	CS		PSM	53	22G	ND
Bentz, 1998 [11]	CS		PSM	60	22G	Yes
Bergeron, 2015 [12]	CS		PSM	1104	ND	Yes
Chen, 2016 [13]	CS		PSM	102	22G	No
de Luna, 2004 [14]	CS	LBP	PSM	67	ND	Yes
Eloubeidi, 2005 [15]	CS		PSM	300	ND	Yes
Eloubeidi, 2006 [16]	CS		PSM	158	22G	ND
Eloubeidi, 2007 [17]	CS		PSM	547	22G	ND
Furuhata, 2017 [18]	CS		PSM	75	22G	Yes
Haba, 2013 [19]	CS		PSM	996	Mixed	Partial
Hashimoto, 2017 [20]	CS	LBP	PSM	126	25G	ND
Hikichi, 2009 [21]	CS		PSM	73	22G	Yes
Ieni, 2015 [22]	CS		PSM	46	22G	ND
Jang, 2017 [23]	CS		PSM	118	22G	ND
Jeong, 2018 [24]	CS		PSM	97	Mixed	No
LeBlanc, 2010 [25]	CS	LBP	PSM	130	22G	Yes
Lee, 2011 [26]	CS	LBP	Mixed	58	Mixed	No
Lee, 2018 (a) [1]		LBP	PSM	48	22G	ND
Lee, 2018 (b) [2]	CS		PSM	73	22G/25G	No
Park, 2017 [27]	CS		PSM	43	Mixed	ND
Pellisé, 2003 [28]	CS		PSM	33	22G	Yes
Qin, 2014 [29]	CS	LBP	PSM	72	22G	No
Ramesh, 2016 [30]	CS		PSM	612	Mixed	Yes
Saxena, 2018 [31]	CS		PSM	147	22G	Yes
Schneider, 2015 [32]	CS		PSM	63	22G	ND
Tada, 2002 [33]	CS		PSM	34	22G	ND
Trisolini, 2017 [34]	CS		PSM	107	25G	No
Uehara, 2011 [35]	CS		PSM	120	Mixed	Yes
Vanbiervliet, 2014 [36]		LBP	PSM	80	22G	ND
Yeon, 2018 [37]	CS	LBP	ND	43	22G	ND

No: number; ROSE: rapid on-site examination; CS: conventional smear; LBP: liquid-based preparation; PSM: pancreatic solid mass; ND: no description.

**Table 2 diagnostics-10-00293-t002:** Diagnostic accuracy in endoscopic ultrasonography-guided fine-needle aspiration according to the cytologic preparation.

	Number of Subsets	Fixed Effect (95% CI)	Heterogeneity Test [*p*-Value]	Random Effect (95% CI)	Egger’s Test [*p*-Value]
Conventional smear	39	0.812 (0.798, 0.825)	<0.001	0.828 (0.798, 0.855)	0.143
Type					
Solid mass	36	0.810 (0.795, 0.823)	<0.001	0.824 (0.792, 0.852)	0.232
Cystic lesion	1	0.800 (0.572, 0.923)	1.000	0.800 (0.572, 0.923)	-
ROSE					
with ROSE	7	0.921 (0.892, 0.943)	0.010	0.928 (0.879, 0.959)	0.079
without ROSE	13	0.777 (0.749, 0.803)	<0.001	0.809 (0.748, 0.858)	0.032
Needle size					
22 gauge	5	0.798 (0.736, 0.848)	0.006	0.808 (0.682, 0.892)	0.557
25 gauge	5	0.779 (0.735, 0.817)	0.008	0.808 (0.720, 0.873)	0.138
Conventional smear					
Slow-pull technique	2	0.729 (0.651, 0.795)	0.047	0.762 (0.577, 0.882)	-
Fanning technique	1	0.588 (0.487, 0.681)	1.000	0.588 (0.487, 0.681)	-
Liquid-based preparation	5	0.867 (0.823, 0.902)	<0.001	0.940 (0.844, 0.978)	0.065
ROSE					
with ROSE	1	0.980 (0.871, 0.997)	1.000	0.980 (0.871, 0.997)	-
without ROSE	1	0.983 (0.888, 0.998)	1.000	0.983 (0.888, 0.998)	-
Needle size					
22 gauge	2	0.983 (0.935, 0.996)	0.810	0.983 (0.935, 0.996)	-
25 gauge	1	0.902 (0.844, 0.940)	1.000	0.902 (0.844, 0.940)	-

CI: Confidence interval; ROSE: rapid on-site examination.

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
