# Peer review of "Comparison between Conventional Smear and Liquid-Based Preparation in Endoscopic Ultrasonography-Fine Needle Aspiration Cytology of Pancreatic Lesions"

_diagnostics, 2020, doi:10.3390/diagnostics10050293_

Round 1
Reviewer 1 Report
The article represents the first meta-analysis in the field. I just suggest to consider in the analysis or at least in the discussion, the impact of the needle size and design on the final results
Author Response
We investigated the impact of needle size on diagnostic accuracy dividing into 22- and 25-gauges. In conventional smear (CS), needle size had no impact on diagnostic accuracy (0.808, 95% CI, 0.682-0.892 vs. 0.808, 95% CI, 0.720-0.873; Table 2). The diagnostic accuracy of LBP with 22-gauge was higher than that of LBP with 25-gauge (0.983, 95% CI, 0.935-0.996 vs. 0.902, 95% CI, 0.844-0.940; Table 2).
The pooled sensitivities of 22- and 25-gauges of CS were 0.89 (95% CI, 0.88-0.91) and 0.78 (95% CI, 0.71-0.84), respectively. The pooled specificities were 0.93 (95% CI, 0.89-0.95) and 1.00 (95% CI, 0.87-1.00), respectively. The diagnostic odds ratios were 91.24 (95% CI, 36.88-225.75) and 120.79 (95% CI, 15.23-957.83) in 22- and 25-gauges of CS, respectively (data not shown). However, diagnostic test accuracy review for comparison between 22- and 25-gauges in LBP could not be performed due to insufficient information.
We added the results and the limitation in the revised manuscript.
Reviewer 2 Report
This study is meta-analysis and DTA review to compare EUS-FNA of pancreatic lesions with conventional smear and LBC. There are some problems to be improved as follows.
Major
- In this paper, the diagnostic accuracy of CS was higher than LBP. However, in table1, it seems that there are more previous papers on CS than those on LBP. The diagnostic ability of EUS-FNA is improving year by year by improving the puncture method (e.g. slow pull method or funning method) and puncture needle (e.g. FNB needle). Is it possible that the difference between these methods influences this result?
- This paper stated that the impact of ROSE might be lower in LBP than in CS. However, one of the merits of performing ROSE is that it can be determined whether an appropriate sample has been collected during EUS-FNA. Why does LBP not contribute to the improvement of diagnostic ability?
- While diagnostic accuracy was higher in LBP than in CS. But, in DTA review, the sensitivity was lower in LBP than in CS and the specificity is similar. What causes this discrepancy?
Author Response
1.In this paper, the diagnostic accuracy of CS was higher than LBP. However, in table1, it seems that there are more previous papers on CS than those on LBP. The diagnostic ability of EUS-FNA is improving year by year by improving the puncture method (e.g. slow pull method or funning method) and puncture needle (e.g. FNB needle). Is it possible that the difference between these methods influences this result?
Response:
As pointed out, the subgroup analysis based on puncture method was performed in CS group, but not in LBP group. The results showed in Supplementary Table 1. We added the results in the revised manuscript.
|
Supplementary Table 1. Diagnostic accuracy in endoscopic ultrasonography-guided fine-needle aspiration according to the puncture methods |
|||||
|
Number of subsets |
Fixed effect [95% CI] |
Heterogeneity test [P-value] |
Random effect [95% CI] |
Egger’s Test [P-value] |
|
|
Conventional smear |
|||||
|
Slow-pull technique |
2 |
0.729 [0.651, 0.795] |
0.047 |
0.762 [0.577, 0.882] |
- |
|
Fanning technique |
1 |
0.588 [0.487, 0.681] |
1.000 |
0.588 [0.487, 0.681] |
- |
|
CI, Confidence interval |
|||||
However, the subgroup analysis based on puncture needle types could not be performed due to insufficient information. We added this limitation in the revised manuscript.
2.This paper stated that the impact of ROSE might be lower in LBP than in CS. However, one of the merits of performing ROSE is that it can be determined whether an appropriate sample has been collected during EUS-FNA. Why does LBP not contribute to the improvement of diagnostic ability?
Response:
First of all, the difference between CS and LBP should be understood. The process of evaluating sample adequacy by ROSE is the same as creating pathologic slides in CS. However, samples from ROSE are different from samples for LBP. In other words, slides by ROSE are different from slides by LBP. Applying ROSE in LBP can be considered to evaluate both LBP and CS slides. Thus, because this is a pure LBP, it is unable to assess any impact on the diagnostic ability of ROSE in LBP. In the current study, in LBP, the application of ROSE has no impact on the diagnostic accuracy of EUS-FNAC (0.980, 95% CI, 0.871-0.997 vs. 0.983, 95% CI, 0.888-0.998).
We added the comment for the role of ROSE in LBP.
3.While diagnostic accuracy was higher in LBP than in CS. But, in DTA review, the sensitivity was lower in LBP than in CS and the specificity is similar. What causes this discrepancy?
Response:
We described the diagnostic accuracy of CS with ROSE and without ROSE in Table 2. The diagnostic accuracies of CS with ROSE and without ROSE were 0.928 (95% CI, 0.879-0.959) and 0.809 (95% CI, 0.748-0.858), respectively. There was no difference in diagnostic accuracy between CS with ROSE and LBP (0.928 vs. 0.940). In the subgroup analysis of DTA review, diagnostic test accuracy of CS with ROSE and LBP was higher than that of CS without ROSE (Supplementary Table 2).
|
Supplementary Table 2. Sensitivity, specificity, diagnostic odds ratio and area under curve of summary receiver operation characteristics curve in endoscopic sonography-guided fine-needle aspiration cytology according to the cytologic preparation. |
|||||
|
Comparison |
Included studies |
Sensitivity (%) [95% CI] |
Specificity (%) [95% CI] |
Diagnostic OR [95% CI] |
AUC on SROC |
|
Conventional smear |
|||||
|
with ROSE |
14 |
0.93 [0.88, 0.96] |
0.94 [0.86, 0.98] |
102.50 [39.24, 267.72] |
0.952 |
|
without ROSE |
5 |
0.84 [0.67, 0.93] |
0.95 [0.61, 1.00] |
24.42 [8.11, 73.47] |
0.884 |
|
Liquid-based preparation |
9 |
0.81 [0.70, 0.89] |
1.00 [0.02, 1.00] |
57.21 [23.61, 138.64] |
0.939 |
|
CI, Confidence interval; OR, Odds ratio; AUC, Area under curve; SROC, summary receiver operating characteristic; FNAC, fine-needle aspiration cytology; FNB, fine-needle aspiration biopsy; ROSE, rapid on-site cytology examination |
|||||
In comparison between CS and LBP, diagnostic test accuracy was slightly higher in CS with ROSE than in LBP, as pointed out by the reviewer. This discrepancy could be caused by the difference in included studies between diagnostic accuracy and DTA review. Thus, we concluded that CS with ROSE and LBP had the highest diagnostic accuracy in EUS-FNAC of pancreatic lesions.
Round 2
Reviewer 2 Report
Thank you for your reply.
I was satisfied with the reply.There are no additional comments.